# Correlations among the nursing work environment, traumatic stress, and professional quality of life in Chinese midwives: A cross-sectional study

Xiaofei Zhang[1], Bing Lun[2], Haojie Ge[1], Lixia Qu[1]*

1 School of Nursing and Health, Zhengzhou University, Zhengzhou, Henan,China, 2 Delivery Room, Maternal and Child Health Hospital of Henan Province, Zhengzhou, Henan, China

* qulixia@zzu.edu.cn

## Abstract

### Background

Midwives work in a high-stress, high-risk, and high-intensity delivery room environment, which exposes them to significant emotional challenges. Understanding the factors influencing midwives' professional quality of life (ProQoL) is crucial for maintaining their well-being. Although the nursing work environment plays a significant role in ProQoL, a gap in understanding how the nursing work environment and traumatic stress affect midwives' ProQoL remains, especially in Chinese midwives.

### Aim

The purpose of this study was to identify how the nursing work environment and traumatic stress are related to ProQoL in Chinese midwives.

### Methods

An online questionnaire was administered to 232 midwives working in the delivery room of 59 hospitals in Henan Province, China. The participants were selected via a convenience sampling approach between November and December 2023. The data collection tools used were the Demographic and professional characteristics Questionnaire, Traumatic Stress Scale for Midwives (TSSM) (consisting of frequency and impact), Nursing Work Environment Scale (NWES), and ProQoL (consisting compassion satisfaction, burnout, and secondary traumatic stress). The data were analyzed via the Mann-Whitney U test, the Kruskal-Wallis H test, Spearman's correlation, and multiple linear regression.

**Data availability statement:** All relevant data are within the paper and its Supporting information files.

**Funding:** This research was supported by Henan Provincial Health Commission (LHGJ20220551), Henan Provincial Health Commission (SBGJ202103077), Henan Federation of Social Sciences (SKL-2024-325). The funders had no role in study design, data collection and analysis, decision to publish, or preparation of the manuscript.

**Competing interests:** The authors have declared that no competing interests exist.

## Results

The study revealed that midwives reported moderate levels of compassion satisfaction ($35.18 \pm 7.703$) and burnout ($25.33 \pm 4.334$), alongside a low level of secondary traumatic stress ($21.50 \pm 5.464$). Results showed that the nursing work environment was positively correlated with compassion satisfaction ($r = 0.610$) and negatively correlated with burnout ($r = -0.390$) and secondary traumatic stress ($r = -0.296$). Midwives' scores on the frequency and impact of traumatic stress were positively related to burnout ($r = 0.254$, $r = 0.452$) and secondary traumatic stress ($r = 0.281$, $r = 0.380$) but negatively related to compassion satisfaction ($r = -0.145$, $r = -0.383$). Multiple regression analysis results revealed that the nursing work environment, the impact of traumatic stress, major shifts, health condition and the frequency of traumatic stress predicted compassion satisfaction. The impact of traumatic stress, health condition, and the nursing work environment predicted burnout. The impact of traumatic stress, the nursing work environment and frequency of night shifts per month predicted secondary traumatic stress.

## Conclusions

The associations we identified among the nursing work environment, traumatic stress, and ProQoL suggest the potential importance of implementing a supportive nursing work environment and developing strategies such as trauma-informed care education and trauma management for midwives. These strategies are vital in improving midwives' ProQoL, thereby promoting their health and well-being.

## Introduction

Midwifery is an emotionally demanding profession, as the delivery room constitutes a high-stress, high-risk, and high-intensity environment [1]. As key professionals in the field of maternal and newborn health, midwives support women during their transition to motherhood [2]. In this role, midwives not only share moments of joy with childbearing women but also witness trauma and loss. The close relationship between midwives and childbearing women is a central feature of midwifery practices [3], serving as a significant source of professional fulfillment. However, midwives face psychological stress from traumatic birth events, including maternal death, fetal or neonatal death, obstetric emergencies, and postpartum complications [4]. Traumatic birth events in the delivery room may result from either direct involvement in emergency situations or indirect witnessing of adverse events [5]. According to the literature, it is common for midwives to witness traumatic birth events. Studies have revealed that 94% of midwives in Israel [6] and 98.2% of midwives in Italy [4] have witnessed traumatic birth events. Prolonged exposure to traumatic birth events may adversely affect midwives' professional quality of life (ProQoL) and mental health.

ProQoL reflects an individual's emotional and psychological well-being derived from their caregiving role [7]. It consists of three key components: compassion

satisfaction, burnout and secondary traumatic stress [7]. Burnout is characterized by feelings of hopelessness and difficulty in efficiently managing one's job [7]. It can have a series of negative impacts on patients, medical institutions, the nursing profession, and the individual physical and mental health of nurses [8]. Secondary traumatic stress refers to work-related exposure to individuals who have experienced traumatic events [7]; it is characterized by insomnia, fear, nightmares, and avoidant behavior [9]. The pressure caused by such trauma increases instances of malpractice, leads to a loss of motivation, and reduces compassion in the nursing process [10]. In addition, health care professionals can also derive pleasure from helping patients; such pleasure is called compassion satisfaction [7]. Compassion satisfaction reflect positive aspect of ProQoL, burnout and secondary traumatic stress reflect negative aspect of ProQoL.

Job demand-resources theory holds that the dynamic balance between demands and resources in the work environment determines employees' occupational health and work outcomes [11]. The job of a midwife is characterized by high emotional demands and high exposure to trauma, and a good working environment can provide better work resources, enhance the emotional recovery ability of midwives, and thereby promote ProQoL. The nursing work environment encompasses the organizational features of the workplace that either facilitate or restrict professional nursing practice [12]. Prior studies have demonstrated that the nursing work environment is related to nurses' well-being, perceived patient-centered care, patient outcomes, and quality of care [13–15]. The supportive work environment, characterized by professional development, collaboration, professional autonomy and values, and meaningful recognition and acknowledgment, can be regarded as a motivation factor and is associated with increased job satisfaction and low levels of burnout and secondary traumatic stress [14,16–19]. Conversely, overburdened and unsupported environments exacerbate negative emotions and decrease ProQoL [20–22]. Thus, the investigation of nursing work environment and related influencing factors is warranted to improve ProQoL among the nurse population.

Midwives, who are essential to the care of mothers and newborns, face unique challenges. They must face and resolve many birth-related situations independently. Traumatic stress caused by complicated delivery makes them face emotional challenges such as psychological distress. In China, compounding these challenges is the lack of human resources for midwifery [23] and the recent introduction of the "three-child" policy in 2021 [24], "three-child" policy has increased the birth rate among older mothers, thereby increasing the complexity of deliveries and potentially elevating midwives' exposure to traumatic events; moreover, the personalized needs of patients are also increasing. These factors create a pressing need to understand the determinants of ProQoL in Chinese midwives and develop targeted interventions to support their well-being. However, there is a paucity of research examining the relationships among the nursing work environment, traumatic stress, and ProQoL in Chinese midwives.

The aim of this study was to fill this gap by examining these associations in the context of Chinese midwives, with the goal of informing effective interventions that improve midwives' ProQoL, and promote their well-being, ultimately promoting the quality of care for childbearing women.

## Methods

### Study design

A cross-sectional, correlational survey of midwives was conducted.

### Setting and participants

Data were collected from a voluntary convenience sample of midwives from 59 hospitals in Henan Province, China, from November to December 2023. The inclusion criteria were as follows: 1) working as a midwife with at least one year of experience in the delivery room and 2) registered nurses who also hold midwifery endorsements. The exclusion criteria were as follows: 1) midwives who were on leave, such as maternity leave or career breaks and 2) nursing students interning in the delivery room. The strengthening the reporting of observational studies in epidemiology (STROBE) criteria [25] were followed in the conduct and reporting of this study.

G*Power 3.1 software was employed for sample size calculation [26]. The parameters were set as follows: an effect size of d = 0.15, an alpha level of 0.05, and a power level of 0.95. The findings revealed a minimum sample size of 213. Our sample size met this requirement.

## Data collection process

We published our research objectives and content in a WeChat group (created by a delivery room head nurse group during a conference), distributed the electronic questionnaire, and invited delivery room head nurses to help distribute the questionnaire to midwives who met the inclusion criteria. The electronic questionnaire was designed by a social media platform called Questionnaire Star (Questionnaire Star is a secure, widely used online platform for data collection in China, with features that limit duplicate responses). Before the questionnaire started, there was a guiding statement indicating that our research was voluntary and confidential, and the first question was,"Do you agree to participate in this study?" After"Yes"was clicked, the participants could start completing the questionnaire. Finally, a total of 247 midwives completed the investigation. Because of low quality responses (for example, all options were consistent, such as 1, 1, 1, 1, 1 or 5, 5, 5, 5, 5), 15 were excluded; ultimately, 232 valid questionnaires were collected, for a valid response rate of 93.93%. Since there are very few male midwives in China, all of the participants in this study were female.

## Ethical considerations

The Department of Ethics Committee, Zhengzhou University, provided ethical approval (No: ZZUIBRB2020−52). Before the study, we obtained informed consent from all midwives. Participants were informed of voluntary participation, they can withdraw at any moment. Additionally, the study was anonymous, their personal information was protected.

## Measurement

1. **The demographic and professional characteristics questionnaire:** this is a self-report tool that includes age, marital status, educational level, hospital level, professional title, employment form, duty, years as a midwife, major shifts, frequency of night shifts per month, personal monthly income, weekly working time, number of deliveries per week on average, health condition, and sleep quality.

2. **The Nursing Work Environment Scale (NWES):** developed by Ye and Shao [27], this 26-item, self-report tool comprises 7 subscales: career development, leadership and management, doctor and nurse relationships, recognition atmosphere, professional autonomy, basic guarantees, and sufficient human resources. A 6-point Likert scale with scores ranging from 1 (strongly disagree) to 6 (strongly agree) was used to score each item. The results range from 26 to 156, and a higher score indicates a better nursing work environment for the individual. In our study, the Cronbach's alpha was 0.962, indicating acceptable reliability. Permission to use this tool was obtained.

3. **The Traumatic Stress Scale for Midwives (TSSM):** designed by Kubota et al. [28], this scale was used to measure midwives' traumatic stress. The Chinese version, translated and revised by Pu et al. [29], was used for this study. The scale has 2 dimensions, including frequency and impact caused by traumatic stress. Each subscale includes 15 identical items. A 4-point Likert scale is used to display the two subscales. The overall score for each subscale ranges from 0 to 45. The frequency subscale has a range of 0 (never) to 3 (always), and the impact subscale has a range of 0 (not at all) to 3 (extremely). A higher frequency score indicates more traumatic stress experience, and a higher impact score indicates greater damage from traumatic stress experience; the Cronbach's alphas were 0.832 and 0.962, respectively, in our study.

4. **Professional Quality of Life Scale (ProQoL):** developed by Stamm [7], and we used the Chinese version [30]. This self-report questionnaire includes 30 items, each of which is rated on a 5-point Likert scale from 1 (never) to 5 (very

often). The scale comprises three subscales: compassion satisfaction, burnout, and secondary traumatic stress, each with 10 items. Items 1, 4, 15, 17, and 29 of the Likert scale are reverse scored. The subscale has a range of 10–50, with a higher score indicating a higher level of regard. According to Stamm (2010), for each of the three subscales, scores can be classified as low (≦22), moderate (23–41), or high (≧42). The ProQoL has shown good reliability, with Cronbach's alphas of 0.87, 0.72, and 0.80 for compassion satisfaction, burnout, and secondary traumatic stress, respectively [31]. The Chinese version also demonstrated good internal consistency [30].

## Data analysis

IBM SPSS 26.0 was used to analyze the data. To determine if the data were normally distributed, the Q-Q test was used. Due to the fact that all the measurement data did not follow a normal distribution, the Mann-Whitney U test and the Kruskal-Wallis H test were employed to analyze participants' compassion satisfaction, burnout, and secondary traumatic stress associated with demographic and professional characteristics, and $p < 0.05$ indicated a statistically significant difference. Spearman's correlation analysis was used to analyze the relationships among participants' nursing work environment, traumatic stress, compassion satisfaction, burnout, and secondary traumatic stress, and a difference of $p < 0.05$ was considered statistically significant. Moreover, multiple linear stepwise regression analysis was used to investigate the impacts of participants' demographic and professional characteristics, traumatic stress, and nursing work environment (as independent variables) on compassion satisfaction, burnout, and secondary traumatic stress (as dependent variables).

## Results

### Demographic and professional characteristics of the participants

The mean age of the participants was 35.69 years (range: 22–54 years, SD = 6.904). Among the 232 participants, 100% were female, 87.1% were married, and 82.3% held a bachelor's degree. Most participants worked in a three-level hospital (66.8%) and had a supervisor nurse title (58.2%). See more demographic details in Table 1.

### Nursing work environment, traumatic stress and professional quality of life measurement scores of the participants

As shown in Table 2, the average score for the nursing work environment was 121.12 (SD = 21.106), and the scores for the frequency of traumatic stress and the impact of traumatic stress were 7.18 (SD = 4.077) and 24.17 (SD = 9.256), respectively.

According to Stamm, results revealed moderate levels of compassion satisfaction (35.18 ± 7.703) and burnout (25.33 ± 4.334), and low levels of secondary traumatic stress (21.50 ± 5.464).

### Correlational results

Table 3 displays the correlations between the primary study variables. The nursing work environment score was positively correlated with compassion satisfaction ($r = 0.610$) but negatively correlated with burnout ($r = -0.390$) and secondary traumatic stress ($r = -0.296$). The results revealed that midwives who reported a healthier nursing work environment reported higher compassion satisfaction levels, lower burnout levels, and lower secondary traumatic stress levels. Midwives' scores on the frequency and impact of traumatic stress were positively related to burnout ($r = 0.254$, $r = 0.452$) and secondary traumatic stress ($r = 0.281$, $r = 0.380$) but negatively related to compassion satisfaction ($r = -0.145$, $r = -0.383$).

### Factors associated with participants' compassion satisfaction, burnout, and secondary traumatic stress

The nonparametric test results are shown in Table 1. Hospital level, years as a midwife, major shifts, personal monthly income, weekly working time, health condition, and sleep quality had different scores for compassion satisfaction. Health

**Table 1. Characteristics of the participants and univariate analyses of the demographic and professional factors associated with compassion satisfaction, burnout, and secondary traumatic stress (N = 232).**

| Characteristics | Category | n (%) | Compassion satisfaction | | Burnout | | Secondary traumatic stress | |
|---|---|---|---|---|---|---|---|---|
| | | | Mean ± SD | Z/F (*p Value*) | Mean ± SD | Z/F (*p Value*) | Mean ± SD | Z/F (*p Value*) |
| **Ages (in years)** | ≦30 | 59 (25.4) | 35.54 ± 7.901 | 0.930 | 25.10 ± 4.521 | 1.204 | 21.15 ± 5.965 | 0.341 |
| | 31-40 | 120 (51.7) | 34.61 ± 7.995 | (0.628) | 25.61 ± 4.133 | (0.548) | 21.56 ± 5.203 | (0.843) |
| | ≧41 | 53 (22.8) | 36.08 ± 6.782 | | 24.94 ± 4.601 | | 21.75 ± 5.547 | |
| **Marital status** | Single | 24 (10.3) | 35.79 ± 7.616 | 2.461 | 24.96 ± 5.137 | 1.226 | 20.33 ± 5.903 | 2.240 |
| | Married | 202 (87.1) | 34.98 ± 7.730 | (0.292) | 25.41 ± 4.253 | (0.542) | 21.70 ± 5.448 | (0.326) |
| | Divorced or windowed | 6 (2.6) | 39.67 ± 6.743 | | 24.00 ± 4.050 | | 19.33 ± 3.445 | |
| **Educational level** | Associate degree | 41 (17.7) | 34.54 ± 6.903 | −0.735 | 24.76 ± 3.992 | −0.885 | 21.41 ± 5.749 | −0.051 |
| | Bachelor's degree | 191 (82.3) | 35.32 ± 7.875 | (0.462) | 25.45 ± 4.404 | (0.376) | 21.52 ± 5.416 | (0.959) |
| **Hospital level** | Three | 155 (66.8) | 36.20 ± 7.427 | −2.955 | 24.94 ± 4.180 | −1.826 | 21.32 ± 5.485 | −0.794 |
| | Two | 77 (33.2) | 33.13 ± 7.888 | (0.003) | 26.12 ± 4.554 | (0.068) | 21.87 ± 5.437 | (0.427) |
| **Professional title** | Primary nurse | 14 (6.0) | 34.64 ± 8.705 | 0.508 | 24.43 ± 4.926 | 1.329 | 20.79 ± 4.509 | 6.817 |
| | Senior nurse | 65 (28.0) | 35.02 ± 7.783 | (0.917) | 25.05 ± 4.185 | (0.772) | 20.68 ± 5.745 | (0.078) |
| | Supervisor nurse | 135 (58.2) | 35.16 ± 7.704 | | 25.46 ± 4.244 | | 21.59 ± 5.426 | |
| | Deputy chief nurse | 18 (7.8) | 36.39 ± 7.114 | | 26.06 ± 5.207 | | 24.33 ± 4.715 | |
| **Employment form** | Staffing at public | 71 (30.6) | 35.69 ± 6.863 | 2.226 | 25.00 ± 4.554 | 2.268 | 21.38 ± 5.131 | 0.615 |
| | Personal agency | 116 (50.0) | 35.45 ± 8.080 | (0.527) | 25.59 ± 4.398 | (0.519) | 21.59 ± 5.778 | (0.893) |
| | Contractor | 42 (18.1) | 33.83 ± 8.076 | | 24.98 ± 3.879 | | 21.62 ± 5.277 | |
| | Temporary worker | 3 (1.3) | 31.67 ± 6.658 | | 27.67 ± 2.082 | | 19.00 ± 5.000 | |
| **Duties** | No | 186 (80.2) | 34.65 ± 7.858 | −1.806 | 25.23 ± 4.344 | −0.565 | 21.17 ± 5.609 | −2.001 |
| | Head nurse | 46 (19.8) | 37.33 ± 6.700 | (0.071) | 25.72 ± 4.319 | (0.572) | 22.83 ± 4.654 | (0.045) |
| **Years as a midwife** | 1-5 | 187 (80.6) | 34.58 ± 7.831 | −2.227 | 25.52 ± 4.457 | −1.152 | 21.42 ± 5.458 | −0.433 |
| | ≧6 | 45 (19.4) | 37.69 ± 6.660 | (0.026) | 24.53 ± 3.721 | (0.249) | 21.84 ± 5.535 | (0.665) |
| **Major shifts** | Day shift | 90 (38.8) | 37.61 ± 6.347 | −3.737 | 24.74 ± 3.928 | −1.405 | 21.58 ± 4.972 | −0.437 |
| | Day-Night shift | 142 (61.2) | 33.64 ± 8.102 | (< 0.001) | 25.70 ± 4.548 | (0.160) | 21.45 ± 5.771 | (0.662) |
| **Frequency of night shifts per month** | <5 | 85 (36.6) | 35.40 ± 7.593 | −0.189 | 25.25 ± 4.367 | −0.347 | 22.53 ± 4.755 | −2.426 |
| | ≧6 | 147 (63.4) | 35.05 ± 7.790 | (0.850) | 25.37 ± 4.329 | (0.728) | 20.90 ± 5.766 | (0.015) |
| **Personal monthly Income (in Chinese Yuan)** | ≦5000 | 132 (56.9) | 33.65 ± 7.785 | 13.976 | 25.33 ± 4.533 | 0.663 | 21.02 ± 5.401 | 3.685 |
| | 5001-10000 | 96 (41.4) | 37.15 ± 7.123 | (0.001) | 25.26 ± 4.135 | (0.718) | 22.00 ± 5.483 | (0.158) |
| | ≧10000 | 4 (1.7) | 38.50 ± 8.660 | | 26.75 ± 2.217 | | 25.50 ± 5.745 | |
| **Weekly working time** | ≦50h | 154 (66.4) | 36.00 ± 7.514 | −2.095 | 25.14 ± 4.232 | −0.987 | 21.71 ± 5.195 | −1.032 |
| | >50h | 78 (33.6) | 33.56 ± 7.865 | (0.036) | 25.71 ± 4.533 | (0.323) | 21.08 ± 5.971 | (0.302) |
| **Number of deliveries per week on average** | 1-10 | 153 (65.9) | 34.54 ± 8.096 | −1.819 | 25.30 ± 4.440 | −0.092 | 21.64 ± 5.471 | −0.52 |
| | ≧11 | 79 (34.1) | 36.42 ± 6.759 | (0.069) | 25.38 ± 4.149 | (0.927) | 21.23 ± 5.475 | (0.603) |
| **Heath condition** | Very good | 59 (25.4) | 39.15 ± 5.539 | 24.274 | 23.46 ± 4.606 | 26.623 | 20.80 ± 6.122 | 2.318 |
| | Good | 75 (32.3) | 35.51 ± 7.318 | (< 0.001) | 24.72 ± 3.754 | (< 0.001) | 21.43 ± 4.998 | (0.314) |
| | Not good | 98 (42.2) | 32.54 ± 8.087 | | 26.92 ± 4.040 | | 21.98 ± 5.394 | |
| **Sleep quality** | Very good | 23 (9.9) | 40.65 ± 5.540 | 15.685 | 23.87 ± 4.957 | 12.989 | 19.35 ± 5.951 | 8.422 |
| | Good | 50 (21.6) | 36.16 ± 7.675 | (0.001) | 24.00 ± 3.974 | (0.005) | 20.40 ± 5.660 | (0.038) |
| | Neutral | 95 (40.9) | 34.26 ± 7.552 | | 25.72 ± 4.227 | | 21.92 ± 5.071 | |
| | Not good | 64 (27.6) | 33.81 ± 7.817 | | 26.31 ± 4.238 | | 22.52 ± 5.463 | |

Abbreviations: SD, standard deviation; *p*: p value for the model.

**Table 2. Nursing work environment scale, traumatic stress scale for midwives, and professional quality of life scale scores (N = 232).**

| | Mean ± SD | High, n (%) | Moderate, n (%) | Low, n (%) |
|---|---|---|---|---|
| **Nursing Work Environment Scale** | 121.12 ± 21.106 | | | |
| Career development | 23.80 ± 4.806 | | | |
| Leadership and management | 17.76 ± 4.547 | | | |
| Doctor and nurse relationships | 18.95 ± 3.565 | | | |
| Recognition atmosphere | 15.49 ± 1.980 | | | |
| Professional autonomy | 19.77 ± 3.255 | | | |
| Basic guarantees | 11.50 ± 4.403 | | | |
| Sufficient manpower | 13.85 ± 3.056 | | | |
| **Traumatic Stress Scale for Midwives** | | | | |
| Frequency | 7.18 ± 4.077 | | | |
| Impact | 24.17 ± 9.256 | | | |
| **Professional Quality of Life Scale** | | | | |
| Compassion satisfaction | 35.18 ± 7.703 | 47(20.3) | 171(73.7) | 14(6.0) |
| Burnout | 25.33 ± 4.334 | 0 | 166(71.6) | 66(28.4) |
| Secondary traumatic stress | 21.50 ± 5.464 | 0 | 91(39.2) | 141(60.8) |

Abbreviations: SD, standard deviation.

Note: Professional quality of life scoring system, low (≦22), moderate (23–41) and high (≧42).

**Table 3. Matrix of correlation between the variables of the study (N = 232).**

| | Compassion satisfaction | Burnout | Secondary traumatic stress |
|---|---|---|---|
| **Nursing work environment** | $r = 0.610^{**}$ | $r = -0.390^{**}$ | $r = -0.296^{**}$ |
| **The frequency of traumatic stress** | $r = -0.145^{*}$ | $r = 0.254^{**}$ | $r = 0.281^{**}$ |
| **The impact of traumatic stress** | $r = -0.383^{**}$ | $r = 0.452^{**}$ | $r = 0.380^{**}$ |

Note: $r$, Spearman coefficient; *, Statistically significant at $p < 0.05$; **, Statistically significant at $p < 0.01$.

condition and sleep quality had different scores for burnout. The secondary traumatic stress scores varied according to duties, frequency of night shifts per month, and sleep quality. There was a statistically significant difference ($p < 0.05$). Next, we used compassion satisfaction, burnout, and secondary traumatic stress as dependent variables. For the independent variables, we chose those that were statistically significant ($p < 0.05$) in the univariate analysis and ($p < 0.05$) in the Spearman's correlation analysis for the multiple linear regression analysis. Table 4 presents the multiple regression results. The nursing work environment, the impact of traumatic stress, major shifts, health condition, and the frequency of traumatic stress had statistically important effects on compassion satisfaction, and the nursing work environment, the impact of traumatic stress and the frequency of traumatic stress accounted for 51.0%, 25.2% and 12.6%, respectively, of the variance in compassion satisfaction. For burnout, three variables (the impact of traumatic stress, health condition, and the nursing work environment) were statistically significant predictors, and the nursing work environment and the impact of traumatic stress explained 19.4% and 38.8%, respectively, of the variance in burnout. Moreover, the impact of traumatic stress, the nursing work environment, and the frequency of night shifts per month had a statistically significant effect on secondary traumatic stress, and the nursing work environment and the impact of traumatic stress accounted for 17.2% and 33.8%, respectively, of the variance in secondary traumatic stress. Moreover, the variance inflation factor (VIF) of all independent variables was below 5 (range: 1.094–1.351), indicating that there was no serious multicollinearity problem.

**Table 4. Multiple linear regression analysis for the impact of characteristics of univariate analysis on compassion satisfaction, burnout, and secondary traumatic stress (N = 232).**

| Variables | B | SE | β | t | p Value | 95% CI (LL, UL) | VIF |
|---|---|---|---|---|---|---|---|
| **Compassion satisfaction** | 21.616 | 3.781 | | | | | |
| Nursing work environment | 0.186 | 0.020 | 0.510 | 9.099 | 0.000 | (0.146,0.226) | 1.351 |
| The impact of traumatic stress | −0.210 | 0.046 | −0.252 | −4.566 | 0.000 | (−0.300, −0.119) | 1.310 |
| Major shifts | −1.767 | 0.795 | −0.112 | −2.222 | 0.027 | (−3.334, −0.200) | 1.094 |
| Heath condition | −1.276 | 0.495 | −0.134 | −2.580 | 0.011 | (−2.251, −0.302) | 1.156 |
| The frequency of traumatic stress | 0.239 | 0.105 | 0.126 | 2.280 | 0.024 | (0.032, 0.445) | 1.325 |
| **Burnout** | 23.491 | 2.085 | | | | | |
| The impact of traumatic stress | 0.182 | 0.027 | 0.388 | 6.640 | 0.000 | (0.128,0.236) | 1.170 |
| Heath condition | 1.042 | 0.307 | 0.194 | 3.391 | 0.001 | (0.436,1.647) | 1.121 |
| Nursing work environment | −0.040 | 0.012 | −0.194 | −3.197 | 0.002 | (−0.064,-0.015) | 1.256 |
| **Secondary traumatic stress** | 24.431 | 2.812 | | | | | |
| The impact of traumatic stress | 0.199 | 0.038 | 0.338 | 5.298 | 0.000 | (0.125,0.274) | 1.170 |
| Nursing work environment | −0.044 | 0.016 | −0.172 | −2.697 | 0.008 | (−0.077, −0.012) | 1.167 |
| Frequency of night shifts per month | −1.449 | 0.670 | −0.128 | −2.164 | 0.031 | (−2.769, −0.130) | 1.008 |

**Compassion satisfaction:** $R^2 = 0.475$ Adjusted $R^2 = 0.464$ F = 40.926 $p = 0.000$

**Burnout:** $R^2 = 0.334$ Adjusted $R^2 = 0.326$ F = 38.192 $p = 0.000$

**Secondary traumatic stress:** $R^2 = 0.207$ Adjusted $R^2 = 0.197$ F = 19.894 $p = 0.000$

Abbreviations: B, Unstandardized Coefficients; SE, Standard error; β, Standardized Coefficients; t, t-test of significance; CI, Confidence interval; LL, Lower limit; UL, Upper Limit; VIF, Variance inflation factor; $R^2$, Coefficient of determination; F, f value for the model; $p$: p value for the model.

## Discussion

This study investigated the status of ProQoL in Chinese midwives, as well as the associations among the nursing work environment, traumatic stress, and ProQoL. In our study, compassion satisfaction and burnout levels were moderate, whereas secondary traumatic stress levels were low (according to the norms provided by Stamm). Compared with those reported in a survey of Italian midwives [4], the levels of compassion satisfaction, burnout, and secondary traumatic stress among Chinese midwives in this study were comparable. Compared with a study of midwifery students at a university in northwest England [32], our study revealed similar levels of secondary traumatic stress, lower compassion satisfaction, and greater burnout. These variations may be attributed to differences in health care systems, cultures, sample sizes, work environments and study methodologies. Notably, China's hierarchical health care system may shape midwives' ProQoL. Unlike in Western contexts with greater midwifery autonomy, Chinese midwives operate within an obstetrician-led framework, potentially limiting decision-making authority and increasing job stress while reducing professional fulfillment factors linked to compassion satisfaction and burnout [33]. Although high workloads and standardized processes may mitigate secondary traumatic stress, they may also reduce emotional engagement, differentially impacting ProQoL compared with more autonomous settings.

The ProQoL of midwives are influenced by multiple factors. Multiple linear regression analysis revealed that major shifts and health condition were important influencing factors of compassion satisfaction. Additionally, health condition was identified as a key factor influencing burnout, while the frequency of night shifts per month emerged as a significant predictor of secondary traumatic stress. The findings of this survey indicate that better health conditions positively influence compassion satisfaction while negatively affecting burnout, which is consistent with other studies [34,35]. Midwives in poor health condition may lack the energy and motivation required for their demanding roles, leading to increased burnout. Additionally, midwives in better health condition are better equipped to provide high-quality care, which enhances

their compassion satisfaction when helping others [35,36]. This study also revealed that midwives working on a day-night shift reported lower compassion satisfaction than those working on a day shift did, which aligns with previous research [37]. Disrupted biological rhythms, sleep deprivation, and fatigue associated with shift work can impair midwives' ability to empathize and derive satisfaction from helping child-bearing women. Additionally, the frequency of night shifts can influence secondary traumatic stress; however, a contrasting study revealed no significant associations between these factors [38]. Frequent night shifts may disrupt social life and sleep patterns; when facing trauma, individuals may be more stressed. Therefore, nursing managers should be aware of midwives' health condition and consider implementing more flexible and supportive scheduling practices to mitigate these effects.

The ProQoL of midwives was correlated with traumatic stress. Multiple regression models revealed that traumatic stress is an important predictor of compassion satisfaction, burnout, and secondary traumatic stress. Most previous studies evaluated exposure to trauma and ProQoL [4,6,39]. In contrast to secondary traumatic stress, traumatic stress is less severe [40]. Traumatic stress in midwives reflects the psychological distress triggered by directly encountering or witnessing obstetric emergencies [30,41], which is a subjective psychological response caused by objective experiences. Secondary traumatic stress is an emotional and psychological response indirectly generated by exposure to the trauma of others [7,42]. Traumatic stress symptoms could serve as a prodromal marker for psychological conditions such as secondary traumatic stress [30,41]. The intimate relationship with child-bearing women is a double-edged sword [6]; such emotional relationships can increase job satisfaction for midwives while also exposing them to traumatic birth events. Thus, frequent witnessing of, and being affected by, traumatic birth events may influence the job satisfaction of midwives and influence the development of compassion satisfaction. Additionally, the impact of traumatic stress, rather than its frequency, emerged as a stronger predictor of burnout. The frequency of traumatic stress did not seem to reflect a stronger impact [6]. Burnout is characterized by elevated levels of emotional exhaustion and is not caused by trauma itself but rather by chronic stress [10]. The greater the emotional relationship with child-bearing women, the greater the impact of trauma has, and a strong sense of responsibility makes midwives feel remorse and guilt, which contributes to burnout [30]. Moreover, the impact of traumatic stress is also an influencing factor of secondary traumatic stress, which means identifying the effects of trauma early and reducing the occurrence of secondary traumatic stress. Hence, it is necessary to develop effective strategies for coping with trauma, such as trauma management and trauma-informed care education (such as structured debriefing sessions following traumatic deliveries, resilience training programs, and routine psychological support) for midwives [43], to improve their ProQoL.

The ProQoL of midwives are linked to the nursing work environment. The regression analysis revealed that the nursing work environment significantly influences compassion satisfaction, burnout, and secondary traumatic stress, corroborating earlier studies [44,45]. A healthier nursing work environment enhances midwives' ProQoL. Career development can help midwives achieve personal and professional growth, enhance job satisfaction, and better manage emotional stress in their work, thus avoiding emotional exhaustion; this shows that nurse managers should establish a platform for midwives' professional development, including training, further study, and career advancement. The characteristics of effective leadership and management include encouragement, patient listening, and providing feedback. It is important to continuously improve the nursing management evaluation system. The doctor-and-nurse relationship is the provision of medical care for patients by nurses and doctors through open communication and coordination under the promise of equality, autonomy, and mutual respect [46]. Good relationships provide emotional support for midwives, helping them cope with work-related stress. Hospital organizations should promote collaboration and support between doctors and midwives through training, communication, and team-building activities. Meaningful recognition may come in various forms, including affirmations by patients and their families, affirmations by other medical staff, satisfactory pay, and value from work, which can be empowering tools to promote retention and reduce turnover [47]. Hospitals should recognize the work performance of midwives through recognition, awards, and performance feedback. Professional autonomy refers to independently making clinical decisions and taking responsibility for one's own professional behavior. Job autonomy enhances

work engagement and positively affects job satisfaction [48], thereby improving midwives' ProQoL. Professional autonomy can be enhanced by increasing public awareness of midwives' roles and competencies, as well as by recognizing their authority [49]. Basic guarantees, including salary, benefits, and leave policies, directly impact the economic well-being and career stability of midwives. These guarantees enhance their sense of belonging, reduce burnout, and serve as a buffer against excessive workload and emotional pressure. Therefore, it is important to optimize salary structures, enhance welfare plans, and implement reasonable vacation policies. Sufficient human resources mean a manageable workload. When faced with an unmanageable workload, individuals often experience significant time pressure, which can lead to a sense of overload, further exacerbating the risk of burnout and secondary traumatic stress [50]. Thus, it is particularly important to optimize human resources.

In conclusion, the findings have significant implications for nursing management and policy development. Interventions aimed at improving midwives' health, optimizing shift schedules, and providing trauma-informed care training and effective trauma management are crucial for improving ProQoL. Additionally, fostering a supportive work environment through effective leadership, adequate staffing, and meaningful recognition can enhance ProQoL. Future studies should employ longitudinal designs, qualitative study and diverse populations to thoroughly investigate the long-term impacts of traumatic stress and work environments on midwives' ProQoL.

### Limitations

First, the research's cross-sectional design limits the inferences that may be made regarding ProQoL, and prohibits inference of causality. Second, due to the use of multiple self-report questionnaires, there may be bias in the results, such as participants considering the risk of social expectation bias, particularly underreporting of burnout or secondary traumatic stress. Third, we used a convenience sample, the study was limited to Henan Province, China, so the sample was geographically homogeneous, leading to selection bias; this sampling method highlight the potential overrepresentation of tertiary hospitals (66.8% of the sample), which may not reflect midwifery conditions in smaller or rural hospitals, moreover, this sampling method was potential for selection bias-for instance, more overburdened or burnout midwives were less likely to respond.

### Conclusions

This research provides valuable insight into the nursing work environment, traumatic stress and relationships with ProQoL in midwives. The findings suggest that nurse managers should focus on the physical and psychological well-being of midwives, develop targeted interventions, foster a healthy work environment, and implement trauma management and trauma-informed care education to improve their ProQoL. The findings also provide valuable evidence for midwives' well-being, adding to the limited knowledge in the field.

### Supporting information

**S1 Data. Minimal data set.**
(XLSX)

### Acknowledgments

The authors extend their gratitude to the midwives who participated in this study and to everyone else who contributed.

### Author contributions

**Conceptualization:** xiaofei zhang.

**Data curation:** xiaofei zhang.

**Investigation:** bing lun.

**Methodology:** bing lun, haojie ge.

**Project administration:** haojie ge.

**Resources:** haojie ge, lixia qu.

**Supervision:** lixia qu.

**Validation:** bing lun.

**Writing – original draft:** xiaofei zhang.

**Writing – review & editing:** lixia qu.

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
