## [Decision Letter · Decision Letter 0]

14 Feb 2025

PONE-D-24-58466The Correlation Between Nursing Work Environment, Traumatic Stress and Compassion Fatigue among Chinese Midwives working in the delivery room: A Cross-Sectional StudyPLOS ONE

Dear Dr. qu,

Thank you for submitting your manuscript to PLOS ONE. After careful consideration, we feel that it has merit but does not fully meet PLOS ONE’s publication criteria as it currently stands. Therefore, we invite you to submit a revised version of the manuscript that addresses the points raised during the review process.

We look forward to receiving your revised manuscript.

Kind regards,

Abdelaziz Hendy, PHD

Academic Editor

PLOS ONE

Journal Requirements:

“Henan Provincial Health Commission”

4. In the online submission form, you indicated that “The questionnaire collection platform (Wenjuanxing) in China is used to collect and store the data on a password controlled computer, which can only be accessed by the author of this study.”

5. Please amend either the title on the online submission form (via Edit Submission) or the title in the manuscript so that they are identical.

Reviewers' comments:

Reviewer's Responses to Questions

**Comments to the Author**

1. Is the manuscript technically sound, and do the data support the conclusions?

Reviewer #1: Partly

Reviewer #2: Yes

2. Has the statistical analysis been performed appropriately and rigorously? 

Reviewer #1: Yes

Reviewer #2: Yes

3. Have the authors made all data underlying the findings in their manuscript fully available?

Reviewer #1: Yes

Reviewer #2: Yes

4. Is the manuscript presented in an intelligible fashion and written in standard English?

Reviewer #1: No

Reviewer #2: Yes

5. Review Comments to the Author

Reviewer #1: All comments are included in the manuscript document.

The paper needs English editing.

The title addresses compassion fatigue and traumatic stress in the work environment; however, the whole paper rounds at CS,BO,and STS. Please clarify

Reviewer #2: The study has good methodological quality and is relevant to the health field, contributing to the understanding of the relationship between work environment, traumatic stress and compassion fatigue among midwives. The research is of sufficient quality for publication, provided minor adjustments are made.

The title is clear and informative, reflecting well the main factors analyzed in the study. However, it could be more objective by reducing redundant information, such as repeating the word “work” and specifying the hospital context. Suggestion: The Correlation Between Nursing Work Environment, Traumatic Stress and Compassion Fatigue among Chinese Midwives: A Cross-Sectional Study.

The abstract presents a well-structured overview of the study, including context, objective, methodology, main results and conclusions. However, the contextualization could be more objective, and the results should emphasize more the practical importance of the correlations found, especially on how the work environment influences compassion fatigue.

The introduction sets out the problem well, highlighting the relevance of compassion fatigue among midwives and its impact on the health system. The literature review is well-founded, providing a solid foundation for the study. However, the justification could be more explicit, directly connecting the gap in the literature with the need for this research.

Methodologically, the study is well designed. However, the authors should clarify how many hospitals were initially eligible and how they arrived at the final sample of 59 institutions. Likewise, the estimated population of midwives and how the participants were approached should be better described. Were there any refusals to take part in the study? What was the response rate to the questionnaires? This information is essential for assessing the representativeness of the sample.

The results are presented objectively, with well-organized averages, standard deviations and percentages. The section correctly highlights how the work environment and traumatic stress impact compassion fatigue.

The discussion contextualizes the findings well in relation to the existing literature, explaining similarities and divergences with previous studies. The impact of the work environment on midwives' mental health is well explored, as is the importance of strategies to reduce compassion fatigue. However, it would be advisable to deepen the discussion on practical applications, suggesting concrete measures to mitigate the problem. In addition, the limitations section could be expanded, emphasizing how convenience sampling can influence the results.

6. PLOS authors have the option to publish the peer review history of their article (what does this mean? ). If published, this will include your full peer review and any attached files.

**Do you want your identity to be public for this peer review?** For information about this choice, including consent withdrawal, please see our Privacy Policy .

Reviewer #1: **Yes: ** Rasha Atia Kadri Ibrahim

Reviewer #2: No

---

## [Author Response · Author response to Decision Letter 1]

25 Apr 2025

Point-by-point response to editors and reviewers

Dear editors and reviewers:

On behalf of my co-authors, I would like to express our sincere gratitude for your valuable comments and suggestions regarding our manuscript. We have carefully considered your feedback and have made concerted efforts to incorporate your suggestions into the revised version. Below are your comments along with our corresponding responses:

Response to editors

Point 1:

Your manuscript, "The correlations among the nursing work environment, traumatic stress, compassion fatigue, and compassion satisfaction among Chinese midwives: A cross-sectional study”, has now been assessed.

Response 1: Thank you for your assessment of our manuscript titled "The correlations among the nursing work environment, traumatic stress, compassion fatigue, and compassion satisfaction among Chinese midwives: A cross-sectional study." We greatly appreciate your feedback and the time you and the reviewers have dedicated to evaluating our work.

Response to reviewer 1

Reviewer #1:

All comments are included in the manuscript document.

The paper needs English editing.

The title addresses compassion fatigue and traumatic stress in the work environment; however, the whole paper rounds at CS,BO,and STS. Please clarify

Thank you for your insightful feedback.

1.The paper needs English editing.

Response:We accepted the comment. We used The American Journal Experts (AJE) for English editing.

2.The title addresses compassion fatigue and traumatic stress in the work environment; however, the whole paper rounds at CS,BO,and STS. Please clarify

Response:We accepted the comment, we have changed the title to “The correlations among the nursing work environment, traumatic stress, compassion fatigue, and compassion satisfaction among Chinese midwives: A cross-sectional study”.

Reviewer #2:

The study has good methodological quality and is relevant to the health field, contributing to the understanding of the relationship between work environment, traumatic stress and compassion fatigue among midwives. The research is of sufficient quality for publication, provided minor adjustments are made.

The title is clear and informative, reflecting well the main factors analyzed in the study. However, it could be more objective by reducing redundant information, such as repeating the word “work” and specifying the hospital context. Suggestion: The Correlation Between Nursing Work Environment, Traumatic Stress and Compassion Fatigue among Chinese Midwives: A Cross-Sectional Study.

The abstract presents a well-structured overview of the study, including context, objective, methodology, main results and conclusions. However, the contextualization could be more objective, and the results should emphasize more the practical importance of the correlations found, especially on how the work environment influences compassion fatigue.

The introduction sets out the problem well, highlighting the relevance of compassion fatigue among midwives and its impact on the health system. The literature review is well-founded, providing a solid foundation for the study. However, the justification could be more explicit, directly connecting the gap in the literature with the need for this research.

Methodologically, the study is well designed. However, the authors should clarify how many hospitals were initially eligible and how they arrived at the final sample of 59 institutions. Likewise, the estimated population of midwives and how the participants were approached should be better described. Were there any refusals to take part in the study? What was the response rate to the questionnaires? This information is essential for assessing the representativeness of the sample.

The results are presented objectively, with well-organized averages, standard deviations and percentages. The section correctly highlights how the work environment and traumatic stress impact compassion fatigue.

The discussion contextualizes the findings well in relation to the existing literature, explaining similarities and divergences with previous studies. The impact of the work environment on midwives' mental health is well explored, as is the importance of strategies to reduce compassion fatigue. However, it would be advisable to deepen the discussion on practical applications, suggesting concrete measures to mitigate the problem. In addition, the limitations section could be expanded, emphasizing how convenience sampling can influence the results.

Thank you for your comment.

1.The title is clear and informative, reflecting well the main factors analyzed in the study. However, it could be more objective by reducing redundant information, such as repeating the word “work” and specifying the hospital context. Suggestion: The Correlation Between Nursing Work Environment, Traumatic Stress and Compassion Fatigue among Chinese Midwives: A Cross-Sectional Study.

Response: Thank you for your suggestion!We have deleted some words “work”.

2.The abstract presents a well-structured overview of the study, including context, objective, methodology, main results and conclusions. However, the contextualization could be more objective, and the results should emphasize more the practical importance of the correlations found, especially on how the work environment influences compassion fatigue.

Response: We accepted the comment.We have revised the results in the abstract, particularly strengthening the impact of nursing work environment on compassion fatigue and compassion satisfaction.We appreciate this comment.

3.The introduction sets out the problem well, highlighting the relevance of compassion fatigue among midwives and its impact on the health system. The literature review is well-founded, providing a solid foundation for the study. However, the justification could be more explicit, directly connecting the gap in the literature with the need for this research.

Response: We accepted the comment. We have added gaps in previous research and the need and significance of this study.

4.Methodologically, the study is well designed. However, the authors should clarify how many hospitals were initially eligible and how they arrived at the final sample of 59 institutions. Likewise, the estimated population of midwives and how the participants were approached should be better described. Were there any refusals to take part in the study? What was the response rate to the questionnaires? This information is essential for assessing the representativeness of the sample.

Response:We accepted the comment.We have made detailed revisions to the methodology and made the process of distributing and collecting questionnaires clearer.

5.The discussion contextualizes the findings well in relation to the existing literature, explaining similarities and divergences with previous studies. The impact of the work environment on midwives' mental health is well explored, as is the importance of strategies to reduce compassion fatigue. However, it would be advisable to deepen the discussion on practical applications, suggesting concrete measures to mitigate the problem. In addition, the limitations section could be expanded, emphasizing how convenience sampling can influence the results.

Response: We accepted the comment. We have added discussions on practical applications, proposed specific measures to alleviate these problems, and also modified the limitations section, emphasizing how the convenience sampling can influence the results.We appreciate this comment.

In conclusion, we sincerely appreciate the time and effort you have dedicated to reviewing our manuscript. We hope that the revisions will make it acceptable for publication in “PLOS ONE”. Thank you once again for your valuable feedback!

Best regards,

Lixia Qu

---

## [Decision Letter · Decision Letter 1]

18 Jun 2025

PONE-D-24-58466R1

The correlations among the n ursing w ork e nvironment, t raumatic s tress ,  co mpassion f atigue ,  and c ompassion satisfaction among Chinese m idwives: A   c ross- s ectional s tudy

mrs lixia qu

PLOS ONE

Dear mrs qu,

We are writing to follow up on our previous email regarding your manuscript reassignment. We are pleased to inform you that your manuscript has been assigned to a new Academic Editor.

The new Academic Editor is aware that your manuscript has already experienced a delay, and we will be following up with them to ensure that the rest of the process runs as smoothly as possible.

If you have any further questions, please contact us and we will be happy to help.

Best wishes,

Editor Assignment Team

PLOS ONE

plosone@plos.org

---

## [Author Response · Author response to Decision Letter 2]

4 Jul 2025

Point-by-point response to editors and reviewers

Dear editors and reviewers:

On behalf of my co-authors, I would like to express our sincere gratitude for your valuable comments and suggestions regarding our manuscript. We have carefully considered your feedback and have made concerted efforts to incorporate your suggestions into the revised version. Below are your comments along with our corresponding responses:

Response to editors

Thank you for submitting your manuscript to PLOS ONE. After careful consideration, we feel that it has merit but does not fully meet PLOS ONE’s publication criteria as it currently stands. Therefore, we invite you to submit a revised version of the manuscript that addresses the points raised during the review process.

Response: Thank you for your assessment of our manuscript titled "The correlations among the nursing work environment, traumatic stress, compassion fatigue, and compassion satisfaction among Chinese midwives: A cross-sectional study." We greatly appreciate your feedback and the time you and the reviewers have dedicated to evaluating our work.

Response to reviewer 3

Thank you for the opportunity to review the manuscript titled “The correlations among the nursing work environment, traumatic stress, compassion fatigue, and compassion satisfaction among Chinese midwives: A cross-sectional study.” The study addresses an important gap in the literature by focusing on professional quality of life (ProQoL) in Chinese midwifery, a population often overlooked in global workforce studies. The methodology is appropriate, and the findings have significant implications for both practice and policy. However, several key issues need to be addressed to enhance the clarity, rigor, and impact of the manuscript.

Response: Thank you for your insightful feedback.

1.Title & Abstract: The current title includes repetitive phrasing (“among... among”). A more concise alternative is:

"Correlations of Nursing Work Environment, Traumatic Stress, Compassion Fatigue, and Satisfaction in Chinese Midwives: A Cross-Sectional Study"

Abstract (Results): Please clearly specify the direction of associations, such as:

“The nursing work environment was positively correlated with compassion satisfaction and negatively correlated with burnout and secondary traumatic stress…”

Response: We very much appreciate the time and effort you’ve put into your comments. We agree with the reviewer’s comment concerning this issue. We have taken into account your and Reviewer4’s and AJE’s opinions and changed the title to “Correlations among the Nursing Work environment, Traumatic Stress, and Professional Quality of Life in Chinese midwives: A Cross-Sectional Study” . Moreover, we have revised the (Abstract) results to make the associations direction clearer (lines 35-46).

2.Sampling and Participation: While you report a 26.49% participation rate from head nurses, it is unclear why 71 out of 268 agreed to distribute the survey. Please clarify if non-participating hospitals were systematically different (e.g., rural vs urban, secondary vs tertiary care).

-Response Rate & Bias: A 93.93% valid response rate is commendable, but the manuscript should address the potential for selection bias—for instance, whether more overburdened or burnt-out midwives were less likely to respond.

-Data Collection Tool: Briefly describe Questionnaire Star, e.g.:

“Questionnaire Star is a secure, widely used online platform for data collection in China, with features that limit duplicate responses.”

Response: Thank you for your valuable suggestion! We agree with your comment, in our data collection process, we have provided a clearer description (lines 139-142). Regarding “A 93.93% valid response rate is commendable, but the manuscript should address the potential for selection bias-for instance, whether more overburdened or burnt-out midwives were less likely to respond.” This is indeed a limitation of our research, and we have added it in the limitations section (line 393-395). Moreover, we have added the characteristics of Questionnaire Star to the Data Collection Tool as your suggestion (lines 144-145). We appreciate these comments

3.Results

-Demographics Table (Table 1): Ensure percentages for all categories are included. For instance, if 87.1% are married, what percentage are single, divorced, or widowed?

-Regression Reporting (Table 4): Specify whether standardized β coefficients were used. Clarify whether variance inflation factors (VIFs) were tested to rule out multicollinearity. Include adjusted R² values for all models—not only variance percentages.

Response: We acknowledge your comments very much, we accepted the comment. In table1, 87.1% are married, 10.3% are single, 2.6% are divorced or windowed (In the design questionnaire, these two are one option). Furthermore, in Regression Reporting (Table 4), we use standardized β coefficients, it is in the fourth column of Table 4 and we use abbreviations (β), the full name annotation is provided below the table. Regarding the question “whether variance inflation factors (VIFs) were tested”, we conducted the test according to your suggestion, and the VIF results are shown in Table 4, and an explanation of the VIF results was added in the final section of the results (the variance inflation factor (VIF) of all independent variables was below 5 (range: 1.094-1.351), indicating that there was no serious multicollinearity problem)(lines 264-266). We placed the result of adjusted R²values for all models in Table 4, and we revised it and move it at the bottom of Table 4, making it more clearer. We appreciate these comments.

4. Discussion

-Practical Applications: The discussion would benefit from specific examples of trauma-informed care education, such as:

“Structured debriefing sessions following traumatic deliveries, resilience training programs, and routine psychological support could be feasible interventions.”

-Cultural Context: Include commentary on how China's hierarchical and centralized healthcare system might reduce midwives' professional autonomy compared to Western nations, this is especially important when interpreting ProQoL findings.

-Policy Implications: Elaborate on the “three-child policy” and its impact. For example:

“The policy has increased the birth rate among older mothers, thereby increasing the complexity of deliveries and potentially elevating midwives’ exposure to traumatic events.”

Response: We thank for your constructive criticism, and time spent analyzing this

manuscript.We accepted the comment. Regarding “Practical Applications”, we have supplemented“Structured debriefing sessions following traumatic deliveries, resilience training programs, and routine psychological support could be feasible interventions.”in trauma-informed care education according to your suggestion (lines 335-336). Regarding “Cultural Context”, we have supplement how China's hierarchical and centralized healthcare system influence midwives ProQoL findings in DISCUSSION section according to your suggestion (lines 279-286). Regarding “Policy Implications”, we have supplement“three-child policy”and its impact in INTRODUCTION section according to your suggestion (lines 108-110). We appreciate these comments.

5. Limitations

Expand this section to include the following:

-Cross-sectional design: Acknowledge that the design prohibits inference of causality.

-Convenience sampling: Highlight the potential overrepresentation of tertiary hospitals (66.8% of the sample), which may not reflect midwifery conditions in smaller or rural hospitals.

-Self-reporting bias: Discuss the risk of social desirability bias, particularly underreporting of burnout or traumatic stress.

Response: Thank you for your suggestion! We have revised the limitations according to your suggestions in the section of Cross-sectional design, Convenience sampling and Self-reporting bias. These suggestions are very comprehensive (lines 385-395).

6. References & Tables

-Reference 49: Verify the PMID number and complete any missing data.

-Supporting Tables (1–4): These are labeled in the supplement but must be clearly cited in the main text (e.g., “See demographic details in Table S1”).

Response: We accepted the comment. We have verify the PMID number and complete any missing data. We added supporting tables, and labeled in the supplement and be clearly cited in the main text (lines 219-220).

7.Critical Issues to Address

7.1 In the Introduction, compassion fatigue is defined as consisting of BO + STS, but in the Abstract, CF, BO, and STS are treated separately. Please revise for conceptual consistency across the manuscript.

Response: We appreciate you pointing this out in your comment. We accepted the comment. We have revised the full text to all define professional quality of life and consist of compassion satisfaction, burnout, and secondary traumatic stress.

7.2 Regression Modeling Details: Indicate whether predictors were entered simultaneously or hierarchically in your regression models. Also, ensure all models report adjusted R², not just variance explained.

Response: We accepted the comment. In the regression model, it is entered by stepwise, which we have modified in the Data analysis (line 207.

7.3 Ethical Consent Clarification: The “Yes/No” question format is appropriate but specify whether this implied consent was explicitly approved by the ethics board, and whether written consent was waived.

Response: We acknowledge your comments very much, we accepted the comment. Regarding informed consent, we made additional modifications during the data collection process, described in detail as follows: the first question was set,“Do you agree to participate in this study?”After clicking“Yes”, you can start filling out the questionnaire (lines 147-148).

7.4 In the Aim, revise“nursing working environment”to“nursing work environment.”

Response: Thanks for your comment. We feel sorry for the error, we have revised it (line 20).

7.5 Ensure all abbreviations are defined at first use: BO = Burnout, STS = Secondary Traumatic Stress, CS = Compassion Satisfaction.

Response: We thank the reviewer for the kind comments. Abbreviations throughout the text have been corrected and harmonized.

7.6 Statistical Convention: The use of p < 0.10 for univariate tests is unconventional. Consider reverting to p < 0.05 or provide justification.

Response: Thank you for your suggestion! As you suggested, we have revised p < 0.10 into p < 0.05, and re-counted the multiple linear regressions (line 203, 250).

7.7 References

The reference list includes many regionally relevant studies but would benefit from the inclusion of additional international and contextually aligned literature to strengthen the manuscript’s theoretical foundations and policy discussion. For example, Ayed et al. (2024) examined the relationship between ICU nurses’ quality of life and their work environment, reinforcing the central research premise that institutional conditions critically influence caregiver well being (INQUIRY). Similarly, Batran et al. (2025) explored the perceptions of neonatal nurses working in high acuity environments and found that elements of the work environment significantly shaped their professional quality of life—especially burnout and compassion satisfaction (PLOS ONE). In a related context, Ejheisheh et al. (2025) investigated the interplay between professional values and caring behaviors among ICU nurses in Palestine, offering both theoretical and empirical support for studying emotional resilience and value based care in high pressure healthcare settings (BMC Nursing). The integration of these references would deepen the manuscript’s scholarly grounding and provide relevant international comparisons to contextualize the findings.

Response: We very much appreciate the time and effort you’ve put into your comments. We agree with your comment concerning this issue.We have read this literature carefully, it fits well with the context of our study, it really strengthens our theoretical foundations and policy discussions, and it is very useful for our article, which we have incorporated into our study (references 45, 44,19).

Response to reviewer 4

1.Tittle: the tittle gives multiple concepts making it difficult to differentiate dependent and independent variables I suggest “The correlations between nursing work environment, traumatic stress, and professional quality of life of Chinese midwives: A cross- sectional study”.

professional quality of life covers compassion fatigue, and compassion satisfaction. compassion fatigue however, includes burnout and secondary traumatic stress this need to rethink about having traumatic stress measured twice.

Response: We acknowledge your comments very much, we have taken into account your and Reviewer3's and AJE’S opinions and changed the title to “Correlations among the Nursing Work environment, Traumatic Stress, and Professional Quality of Life in Chinese midwives: A Cross-Sectional Study” . Regarding traumatic stress and secondary traumatic stress, we describe it in the third paragraph of our discussion, for example “In contrast to secondary traumatic stress, traumatic stress is less severe”. Your comment are valuable in improving the quality of our manuscript (lines 312-319).

2.Abstract: the background of the abstract focused mainly on compassion satisfaction and ignored other variables so please give more comprehensive background in the abstract. In results please revised this sentence as it requires clarification (The impact and frequency of traumatic stress accounted for 25.2% and 12.6%, respectively, of compassion satisfaction). Key words require revision to include main study variables. Suggestion (workplace environment, compassion fatigue, compassion satisfaction, midwives, mainland China)

Response: We are very grateful for your comments and thoughtful suggestions. As you suggested, we have revised the background of the abstract (focused on professional quality of life) (lines 12-18), results (described correlation analysis results and multiple regression results) (lines 35-46), and key words (lines 53-54).

3.Introduction: Also, in introduction you started with compassion fatigue, so you need to arrange your ideas along the manuscript starting with the tittle started with the work environment. At the end of your introduction please state the aim clearly.

Response: We are very grateful for your comments on the manuscript. We have reorganized the introductory section to begin with the context in work environment (lines 57-71) and stated the aim more clearly (lines 116-119.

4.Methods: please explain why you place the exclusion criteria, because it is not possible, all midwives should participate or attend continuing education sessions. For measuring sample size, what was the total number of midwives?

Please state the dependent and independent variables.

“With one head nurse from each hospital, and there were 268 head nurses in the group” what do you mean by the group? Please cite the reason where you excluded 15 responses.

The instruments were not arranged properly as the tittle and the remining of the paper.

What is the difference between secondary traumatic stress measure as part of the compassion fatigue and traumatic stress scale for midwives? Please clarify.

Response: Thanks for your valuable suggestions! I have re-described the exclusion criteria, in fact we mainly exclude midwives and who are not in post for various reasons and nursing students interning in the delivery room (lines 129-131).

Regarding “With one head nurse from each hospital, and there were 268 head nurses in the group what do you mean by the group? Please cite the reason where you excluded 15 responses”. we am sorry for our error, I meant Wechat group (line 139), we have revised it. Moreover, we explained the reason of excluding 15 responses in the end of Data collection process (lines 149-152).

Regarding “The instruments were not arranged properly as the tittle and the remining of the paper.” we have rearrange the instruction order as the tittle and the remining of the paper.

Regarding “What is the difference between secondary traumatic stress measure as part of the compassion fatigue and t

---

## [Decision Letter · Decision Letter 2]

6 Jul 2025

C orrelations among  the   nursing work   e nvironment, traumatic stress, and professional quality of life   in Chinese m idwives: A   cross-sectional study

PONE-D-24-58466R2

Dear Dr. QU,

We’re pleased to inform you that your manuscript has been judged scientifically suitable for publication and will be formally accepted for publication once it meets all outstanding technical requirements.

Kind regards,

Majed Sulaiman Alamri, PhD

Academic Editor

PLOS ONE

Additional Editor Comments (optional):

Reviewers' comments:

Reviewer's Responses to Questions

**Comments to the Author**

1. If the authors have adequately addressed your comments raised in a previous round of review and you feel that this manuscript is now acceptable for publication, you may indicate that here to bypass the “Comments to the Author” section, enter your conflict of interest statement in the “Confidential to Editor” section, and submit your "Accept" recommendation.

Reviewer #3: All comments have been addressed

Reviewer #5: All comments have been addressed

2. Is the manuscript technically sound, and do the data support the conclusions?

Reviewer #3: Yes

Reviewer #5: Yes

3. Has the statistical analysis been performed appropriately and rigorously? 

Reviewer #3: Yes

Reviewer #5: Yes

4. Have the authors made all data underlying the findings in their manuscript fully available?

Reviewer #3: Yes

Reviewer #5: Yes

5. Is the manuscript presented in an intelligible fashion and written in standard English?

Reviewer #3: Yes

Reviewer #5: Yes

6. Review Comments to the Author

Reviewer #3: The revisions are comprehensive, and no further changes are needed. The manuscript is ready for acceptance.

Reviewer #5: accepted

no comments

thank you

the authors response to all comments

now it is acceptable to be published

7. PLOS authors have the option to publish the peer review history of their article (what does this mean? ). If published, this will include your full peer review and any attached files.

**Do you want your identity to be public for this peer review?** For information about this choice, including consent withdrawal, please see our Privacy Policy .

Reviewer #3: **Yes: ** Dr Ibrahim Aqtam

Reviewer #5: No

---

## [Editor Report · Acceptance letter]

PONE-D-24-58466R2

PLOS ONE

Dear Dr. qu,

I'm pleased to inform you that your manuscript has been deemed suitable for publication in PLOS ONE. Congratulations! Your manuscript is now being handed over to our production team.

Kind regards,

on behalf of

Prof. Majed Sulaiman Alamri

Academic Editor

PLOS ONE